# Development and validation of a clinical prediction model for in-hospital heart failure risk following PCI in patients with coronary artery disease

Zhenlian Ning[1☺], Bing Li[1☺], Ziming Ning[2], Beili Zhu[2], Mengfan Zhao[1], Bin Huang ●[3*]

**1** The Second Clinical Medical College, Henan University of Chinese Medicine, Zhengzhou, China, **2** The First People's Hospital of Pingdingshan, Pingdingshan, China, **3** Henan Provincial Hospital of Traditional Chinese Medicine, Zhengzhou, China

☺ These authors contributed equally to this work.
* hbfred@sina.com

## Abstract

### Objective

Patients with acute coronary syndrome (ACS) are at increased risk of in-hospital heart failure (HF) following percutaneous coronary intervention (PCI), yet understanding of the associated risk factors is limited. This study aims to identify predictors of in-hospital HF after PCI and to develop and validate a clinical prediction model for the early identification of high-risk patients.

### Methods

We retrospectively analyzed data from the patients hospitalized for ACS who underwent PCI at Henan Provincial Hospital of Traditional Chinese Medicine from 01/01/2019–01/10/2023. Patients were classified into non-HF and HF groups based on the occurrence of heart failure after PCI. LASSO regression and logistic regression were employed to identify potential predictors. The model's diagnostic efficacy was assessed using receiver operating characteristic curves and calibration curves, while decision curve analysis and clinical impact curve were utilized to evaluate clinical benefits.

### Results

A total of 309 patients were included in this study, of whom 79.93% were male, with a mean age of 57.84. Key predictors included New York Heart Association (NYHA) classification, smoking status, right coronary artery occlusion after PCI, left ejection fraction (LVEF), and N-terminal fragment of brain natriuretic peptides. The area under the curve (AUC) was 0.910 (95% CI: 0.868–0.953), indicating strong predictive ability.

**Editor:** Arturo Cesaro, University of Campania Luigi Vanvitelli Department of Translational Medicine: Universita degli Studi della Campania Luigi Vanvitelli Dipartimento di Scienze Mediche Traslazionali, ITALY

**Data availability statement:** All relevant data are within the paper and its Supporting Information files.

**Funding:** The author(s) received no specific funding for this work.

**Competing interests:** The authors have declared that no competing interests exist.

Decision curve analysis and clinical impact curve demonstrated good clinical applicability of the nomogram.

## Conclusion

The identified predictors and the prediction model can be used in identifying high-risk individuals who develop HF hospital admission after PCI, or as a basis for further guiding personalized prevention and treatment.

---

## 1. Introduction

Coronary artery disease (CAD) is a pathological process characterized by atherosclerotic plaque accumulation in the epicardial arteries [1,2]. The dynamic nature of the CAD process is categorized as either acute coronary syndromes (ACS) or chronic coronary syndromes (CCS). ACS is an acute onset of heart disease that can be subdivided into two categories: ST-elevation myocardial infarction (STEMI) and non-ST-elevation ACS (NSTE-ACS). NSTE-ACS includes both non-ST elevated myocardial infarction (NSTEMI) and unstable angina (UA) [3,4]. Despite significant advances in the diagnosis and treatment of ACS, cardiovascular disease remains the leading cause of death worldwide. Each year, more than 7 million people worldwide are diagnosed with ACS, including more than 1 million hospitalizations in the United States alone [5].

Percutaneous coronary intervention (PCI) is the cornerstone treatment for ACS, effectively alleviating myocardial ischemia [6,7]. However, accumulating evidence indicates that up to 20% of patients with acute myocardial infarction (AMI) experience poorer outcomes despite successful PCI [8]. Among these, heart failure following PCI poses a life-threatening risk, often leading to progressive deterioration [9,10]. Particularly in patients with left ventricular dysfunction and multivessel disease, despite improved myocardial perfusion from revascularization, the risk of heart failure remains high [11]. The REVIVED-BCIS2 trial found that PCI did not significantly improve survival in patients with multivessel disease and left ventricular dysfunction [12]. This may be related to the severity of myocardial injury and the completeness of revascularization. Heart failure is a complex clinical syndrome characterized by the heart's inability to pump effectively due to myocardial stiffness or weakness [13,14]. Identifying the factors that influence the development of heart failure in ACS patients after PCI has been a longstanding concern. Although numerous studies have explored this topic, they vary widely in design and inconsistent results, particularly pinpointing the most critical risk factors [15,16]. To date, only the study by Yu et al. [17] has established a predictive model for in-hospital mortality risk in patients with acute ST-elevation myocardial infarction and acute heart failure post-PCI. However, this study did not specifically examine the critical window from post-PCI to the onset of in-hospital heart failure and eventual in-hospital mortality. Early identification of high-risk individuals during this period may help reduce the risk of death.

Therefore, the present study expanded on the potential risk factors based on the previous study. Based on cases from the Henan Provincial Hospital of Traditional Chinese Medicine, it aims to develop a clinical prediction model to identify individuals at risk for in-hospital heart failure post-PCI, providing a scientific basis for establishing a clinical screening tool for heart failure following coronary intervention.

## 2. Materials and methods

### 2.1. Study population and selection criteria

This retrospective study was based on the database of the Chest Pain Centre specializing in patients with ACS at Henan Provincial Traditional Chinese Medicine Hospital. It included patients hospitalized for acute coronary syndrome who underwent PCI from 01/01/2019–01/10/2023, and met the inclusion criteria. Following approval from the Institutional Ethics Committee, data retrieval and analysis took place from 01/06/2024–01/11/2024. Patients completed a registration of basic information and laboratory tests within 24 hours of admission, and echocardiography was performed within the third day after PCI. The study was conducted in accordance with the principles of the Declaration of Helsinki and was approved by the Medical Research Ethics Committee of Henan Provincial Traditional Chinese Medicine Hospital (Approval Number: 2024ZY3053). Due to the retrospective design, written informed consent from participants was not obtained.

**Inclusion criteria were as follows:** (1) Data collected from 01/01/2019–01/10/2023; (2) Patients who met the American College of Cardiology/American Heart Association guidelines for coronary artery disease; (3) Patients who were indicated for PCI treatment. The indications for PCI include patients with confirmed STEMI, as well as those with NSTE-ACS who meet any of the following high-risk criteria for immediate PCI: hemodynamic instability, recurrent or persistent chest pain unresponsive to medical treatment, acute atrial fibrillation presumed to be due to ongoing myocardial ischemia, life-threatening arrhythmias or cardiac arrest, mechanical complications, and dynamic ECG changes indicative of recurrent ischemia [3].

**Exclusion criteria included:** (1) Patients with missing baseline data or clinical information; (2) Patients with malignant tumors, cognitive disorders, severe mental illnesses, or uncontrolled systemic diseases such as primary liver or kidney failure; (3) Patients who underwent imaging only without receiving stenting or balloon angioplasty; (4) Patients who experienced other complications post-PCI (such as bleeding, coronary artery perforation, or stent thrombosis); (5) Patients with a clear history of acute or chronic heart failure due to acute myocarditis, dilated cardiomyopathy, or rheumatic heart disease prior to the procedure.

### 2.2. Assessment of outcome

With reference to the 2021 European Society of Cardiology (ESC) and 2022 American College of Cardiology/American Heart Association (ACC/AHA) consensus guidelines [18,19], the diagnosis of HF requires a combination of symptoms, signs, and objective clinical markers such as ancillary tests, including: (1) the presence of at least one of the typical symptoms including resting or active dyspnea, fluid retention, and a significant decrease in activity tolerance; (2) signs of cardiogenic abnormalities on physical examination, including elevated jugular venous pressure, pulmonary crackles, peripheral edema, or abnormal cardiac auscultation; (3) objective indicators of structural or functional cardiac abnormalities, including echocardiographic evidence of reduced (≤40%), mildly reduced (41–49%), or preserved (≥50%) LVEF with diastolic dysfunction, or elevated natriuretic peptide levels (BNP ≥ 100 pg/mL or NT-proBNP ≥ 300 pg/mL in chronic settings, with a higher threshold in the acute phase such as BNP ≥ 500 pg/mL or NT-proBNP ≥ 1000 pg/mL as per ACC/AHA guidelines); and (4) systematic exclusion of non-cardiac etiologies, such as chronic lung disease, cirrhosis, nephrotic syndrome, abnormal thyroid function, or severe anemia.

Patients who developed heart failure after PCI were classified into heart failure with reduced ejection fraction (HFrEF), heart failure with intermediate ejection fraction (HFmrEF), and heart failure with preserved ejection fraction (HFpEF). The

diagnosis of HFrEF is based on symptoms and/or signs of HF and left ventricular ejection fraction (LVEF) < 40%; the diagnosis of HFmrEF is based on symptoms and/or signs of HF, LVEF = 41–49%, and other evidence of structural heart disease (including increased left atrial (LA) size, left ventricular hypertrophy (LVH) or echocardiographic measures of LV filling) make the diagnosis more likely. HFpEF is diagnosed on the basis of symptoms and/or signs of HF, LVEF ≥ 50%, and objective evidence of cardiac structural and/or functional abnormalities consistent with the presence of LV diastolic dysfunction/raised LV filling pressures, including raised natriuretic peptides. (The greater the number of abnormalities present, the higher the likelihood of HFpEF).

## 2.3. Potential predictors

This study is based on previous research evidence and expert opinions, categorizing the potential predictors of post-PCI heart failure in patients with coronary artery disease into three main categories: patient demographics and medical history, cardiovascular-related clinical data at admission, and laboratory indicators obtained upon admission.

The demographics and medical history of the patients include: age, gender (male, female), smoking and drinking status (yes, no), marital status (married, others), hypertension status (yes, no), diabetes status (yes, no), history of cerebral infarction (yes, no), and prior PCI experience (yes, no).

Cardiovascular-related clinical data at admission included type of ACS (UA, UNSTEMI and STEMI), the number of coronary artery lesions post-PCI (none, single-vessel, multi-vessel), time of patient delay (time from arrival at the hospital to the opening of occluded coronary arteries in ACS patients), occlusion status of the left anterior descending artery (LAD), left circumflex artery (LCX), and right coronary artery (RCA) post-PCI (yes, no), New York Heart Association (NYHA) classification (levels I, II, III, IV), left ventricular ejection fraction (LVEF), left ventricular electrical delay (LVED), and intraoperative heparin dosage.

Laboratory indicators at admission included total cholesterol (TC), triglycerides (TG), low-density lipoprotein (LDL), high-density lipoprotein (HDL), cardiac troponin I (c-TNI), creatine kinase-MB (CK-MB), myoglobin (MYO), NT-pro-BNP, D-dimer, glycated hemoglobin (HbA1c), and urea. These laboratory indicators were classified according to the critical value standards outlined in the supplementary materials (see S1 Table ) into normal, abnormal high, and abnormal low groups. Due to a high proportion of missing data for variables such as lesion length, inflammatory markers, peak troponin concentration, and the number of stents implanted during PCI, these factors were not included in the analysis.

## 2.4. Statistic analysis

All continuous variables were assessed for normality. Those with a normal distribution were presented as mean ± standard deviation (SD), while non-normally distributed variables were expressed as median, minimum and maximum value. Differences between groups were evaluated using the Student *t*-test or the rank sum test, as appropriate. Categorical variables were reported as frequencies and proportions, with differences between groups assessed using the Chi-squared test. To identify key predictors, we employed a LASSO regression model and examined the associations between these predictors and outcomes using a multivariable logistic regression model, without including any interaction terms. Key predictors were further refined based on the Akaike Information Criterion (AIC) and the significance of each variable. We fitted the model not only in the overall population but also within different genders and age subgroups (with age divided by the median) to assess potential bias.

The constructed model was evaluated jointly using the receiver operating characteristic (ROC) curve and the calibration curve. ROC area under the curve (AUC) values of 0.80–0.89 and 0.90 or greater were considered indicative of good and excellent diagnostic performance, respectively [20]. The calibration curve was assessed through subjective evaluation and Brier scores, where scores ranging from 0 to 0.25 indicated better predictive ability, with smaller values representing superior performance [21]. To assess the internal validity of the model, we performed 1,000 bootstrap resamples. In each iteration, a new dataset was generated by sampling with replacement from the original dataset while maintaining the same

sample size. The predictive model was then refitted to each resampled dataset, and AUC and Brier score were calculated. Finally, the average of the 1,000 calculations was reported as the internal validation result.

Based on the identified clinical prediction model, we developed heart failure risk nomograms to identify potentially high-risk individuals in clinical settings. Additionally, we conducted clinical decision curve analysis (DCA) and clinical impact curves (CIC) to evaluate the clinical utility of the model. DCA assesses the risks of undertreatment and overtreatment during patient visits, aiding in treatment selection and decision-making [22]. The multivariate imputation by chained equations method was applied to generate five imputed datasets after 100 iterations to handle a small amount of missing data [23]. In this study, a *P* value of <0.05 was considered statistically significant. All statistical analyses were conducted using SPSS version 26.0 and R version 4.3.0.

## 3. Result

### 3.1. Basic characteristics of the study population

A total of 309 patients were included in this study, of whom 247 (79.93%) were male, with a mean age of 57.84 years (standard deviation = 13.38 years old). Among these patients, 282 (91.26%) were married, 105 (33.98%) were smokers, and 60 (19.42%) were alcohol drinkers. Regarding comorbidities, 165 (53.40%) had hypertension, 81 (26.21%) had diabetes, and 57 (18.45%) had a history of cerebral infarction. Additionally, 28.16% had single coronary artery lesions post-PCI, and 39.81% had multiple lesions post-PCI. The prevalence of heart failure (HF) after PCI was 26.21% (n = 81), with HFpEF, HFrEF, and HFmrEF comprising 70.38%, 14.81%, and 14.81% of HF patients, respectively. Univariate analysis showed that the differences in the distribution of 9 variables, age, gender, RCA occlusion post-PCI, LVEF, cTnI, MYO, NTproBNP, and NYHA, were statistically significant between the patients in the HF group and the Non-HF group (all *P*-value < 0.05, Table 1).

### 3.2. Screening for potential predictors

A total of seven key variables were identified through LASSO regression, including NYHA, smoking, RCA occlusion post-PCI, LVEF, NT-proBNP, HDL, and MYO (Fig 1). Based on the AIC and the significance of results from the multivariable logistic regression model, we further excluded HDL and MYO. The final analysis revealed that NYHA (OR = 9.232, 95%CI = 4.949–19.220), smoking (OR = 2.730, 95%CI = 1.224–6.261), RCA-occlusion (OR = 2.647, 95%CI = 1.094–6.328), LVEF (OR = 3.547, 95%CI = 0.997–12.222), and NT-proBNP (OR = 3.909, 95%CI = 1.735–8.886) were significantly associated with the risk of developing HF after PCI (Table 2 and S2 Table). S3 Table presents the model results across different genders and age subgroups. The findings in male and older age groups were generally consistent with the main analysis, whereas in female and younger age groups, only NYHA remained statistically significant.

### 3.3. Evaluation of clinical prediction model

Fig 2 illustrated the ROC curve (Fig 2A) and the calibration curve (Fig 2B), indicating that the clinical prediction model demonstrates excellent diagnostic efficacy (AUC = 0.910, 95% CI = 0.868–0.953) and predictive power (Brier score = 0.083). The results of the internal validation were generally consistent with those of the primary analysis (AUC = 0.918, Brier score = 0.084).

### 3.4. Application of clinical prediction model

Fig 3A presented the nomogram of this clinical prediction model, clearly delineating the scoring criteria and enhancing clinical utility. To evaluate the potential clinical benefit of the constructed model, we performed decision curve analysis (Fig 3B) and clinical impact curve analysis (Fig 3C). The results from both curves indicated that, as the probability threshold increased, our predictive model could achieve a high rate of clinical benefit in determining whether PCI patients experienced postoperative heart failure compared to interventions based on either all patients or none.

**Table 1.  Univariate analysis of the basic characteristics between the HF group and the Non-HF group (n = 309).**

| Variables | | Non HF group (n = 228) | HF group (n = 81) | t/Z/χ² | P |
|---|---|---|---|---|---|
| Continuous variable | | Mean ±SD/ Median (min, max) | | | |
| Age (years old) | – | 56.45 ± 12.85 | 61.75 ± 14.12 | −3.106 | 0.002 |
| Heparin (U) | – | 8000 (5500, 15000) | 8000 (6000, 20000) | −1.542 | 0.123 |
| Time of patient delay (minutes) | – | 69 (16, 1395) | 74 (25, 294) | −0.916 | 0.36 |
| NYHA | – | 1 (1, 4) | 2 (1, 4) | −12.372 | <0.001 |
| Categorical variable | Level | n (%) | | | |
| Gender | Male | 190 (83.3) | 57 (70.4) | 6.262 | 0.012 |
| | Female | 38 (16.7) | 24 (29.6) | | |
| Hypertension | No | 108 (47.4) | 36 (44.4) | 0.205 | 0.65 |
| | Yes | 120 (52.6) | 45 (55.6) | | |
| Diabetes | No | 170 (74.6) | 58 (71.6) | 0.270 | 0.603 |
| | Yes | 58 (25.4) | 23 (28.4) | | |
| Cerebral infarction | No | 191 (83.8) | 61 (75.3) | 2.846 | 0.092 |
| | Yes | 37 (16.2) | 20 (24.7) | | |
| Prior PCI experience | No | 202 (88.6) | 70 (86.4) | 0.269 | 0.604 |
| | Yes | 26 (11.4) | 11 (13.6) | | |
| Smoking | No | 155 (68.0) | 49 (60.5) | 1.494 | 0.222 |
| | Yes | 73 (32.0) | 32 (39.5) | | |
| Drinking | No | 182 (79.8) | 67 (82.7) | 0.319 | 0.572 |
| | Yes | 46 (20.2) | 14 (17.3) | | |
| Marriage | Married | 209 (91.7) | 73 (90.1) | 0.178 | 0.673 |
| | Others | 19 (8.3) | 8 (9.9) | | |
| Number of coronary artery lesions post-PCI | None | 77 (33.8) | 22 (27.2) | 2.402 | 0.301 |
| | Single | 66 (28.9) | 21 (25.9) | | |
| | Multiple | 85 (37.3) | 38 (46.9) | | |
| LAD occlusion post-PCI | No | 161 (70.6) | 59 (72.8) | 0.144 | 0.813 |
| | Yes | 67 (29.4) | 22 (27.2) | | |
| LCX occlusion post-PCI | No | 170 (74.6) | 53 (65.4) | 2.480 | 0.115 |
| | Yes | 58 (25.4) | 28 (34.6) | | |
| RCA occlusion post-PCI | No | 190 (83.3) | 55 (67.9) | 8.667 | 0.003 |
| | Yes | 38 (16.7) | 26 (32.1) | | |
| HbA1c | Normal | 165 (72.4) | 55 (67.9) | 0.582 | 0.446 |
| | Abnormal high | 63 (27.6) | 26 (32.1) | | |
| TC | Normal | 183 (80.3) | 66 (81.5) | 0.057 | 0.812 |
| | Abnormal high | 45 (19.7) | 15 (18.5) | | |
| TG | Normal | 151 (66.2) | 59 (72.8) | 1.200 | 0.273 |
| | Abnormal high | 77 (33.8) | 22 (27.2) | | |
| LDL | Normal | 202 (88.6) | 72 (88.9) | 0.005 | 0.943 |
| | Abnormal high | 26 (11.4) | 9 (11.1) | | |
| HDL | Normal | 90 (39.5) | 25 (30.9) | 1.896 | 0.169 |
| | Abnormal low | 138 (60.5) | 56 (69.1) | | |
| Urea | Normal | 203 (89.0) | 66 (81.5) | 5.137 | 0.077 |
| | Abnormal low | 8 (3.5) | 2 (2.5) | | |
| | Abnormal high | 17 (7.5) | 13 (16.0) | | |

*(Continued)*

**Table 1.** (Continued)

| Variables | | Non HF group (n = 228) | HF group (n = 81) | t/Z/χ² | P |
|---|---|---|---|---|---|
| LVEF | Normal | 217 (95.2) | 57 (70.4) | 36.613 | <0.001 |
| | Abnormal low | 11 (4.8) | 24 (29.6) | | |
| LVED | Normal | 210 (92.1) | 69 (85.2) | 3.265 | 0.071 |
| | Abnormal high | 18 (7.9) | 12 (14.8) | | |
| c-TnI | Normal | 154 (67.5) | 44 (54.3) | 4.540 | 0.033 |
| | Abnormal high | 74 (32.5) | 37 (45.7) | | |
| MYO | Normal | 157 (68.9) | 37 (45.7) | 13.745 | <0.001 |
| | Abnormal high | 71 (31.1) | 44 (54.3) | | |
| CKMB | Normal | 39 (17.1) | 7 (8.6) | 3.379 | 0.098 |
| | Abnormal high | 189 (82.9) | 74 (91.4) | | |
| NTproBNP | Normal | 183 (80.3) | 27 (33.3) | 60.454 | <0.001 |
| | Abnormal high | 45 (19.7) | 54 (66.7) | | |
| D2 | Normal | 194 (85.1) | 50 (61.7) | 19.633 | <0.001 |
| | Abnormal high | 34 (14.9) | 31 (38.3) | | |
| ACS | UA | 80 (35.1) | 27 (33.3) | 0.159 | 0.924 |
| | UNSTEMI | 30 (13.2) | 10 (12.3) | | |
| | STEMI | 118 (51.8) | 44 (54.3) | | |

Note: ACS, Acute coronary syndrome; c-TNI, cardiac troponin I; CK-MB, creatine kinase-MB; D2, D-dimer; HDL, high-density lipoprotein; HbA1c, glycated hemoglobin; LAD, left anterior descending artery; LCX, left circumflex artery; LDL, low-density lipoprotein; LVEF, left ventricular ejection fraction; LVED, left ventricular electrical delay; MYO, myoglobin; NYHA, New York Heart Association classification; NSTEMI, non-ST-elevation myocardial infarction; RCA, right coronary artery; PCI, Percutaneous coronary intervention; STEMI, ST-elevation myocardial infarction; TC, total cholesterol; TG, triglycerides; UA, unstable angina.

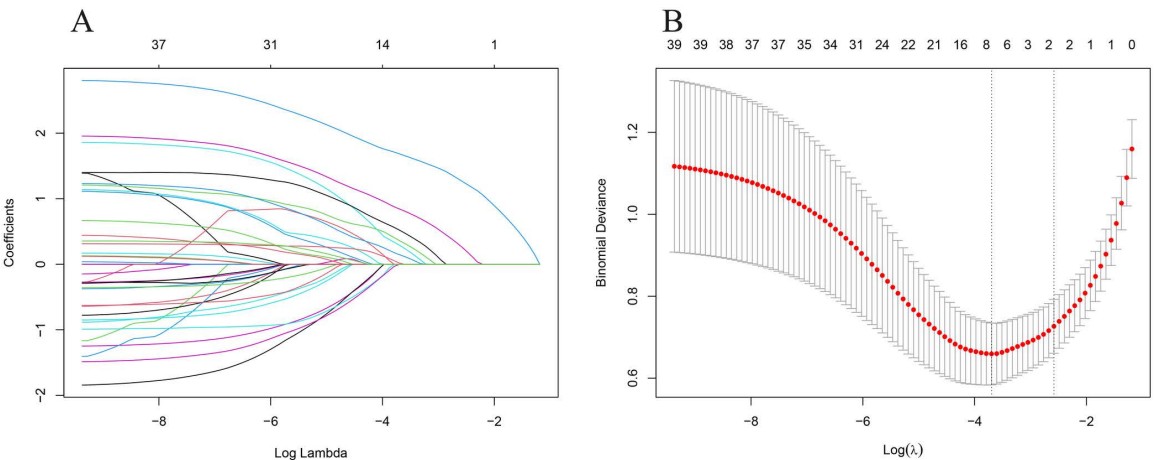

**Fig 1. LASSO regression for variable screening.** (A) LASSO coefficient path; (B) LASSO regularization path.

**Table 2. Multivariable logistic regression analysis of the risk of HF after PCI in patients with ACS.**

| Variables | LASSO regression | | | After model simplification | | |
|---|---|---|---|---|---|---|
| | *OR* | 95%CI | *P* | *OR* | 95%CI | *P* |
| NYHA | 9.201 | 4.827–19.816 | <0.001 | 9.232 | 4.949–19.220 | <0.001 |
| Smoking | 2.576 | 1.138–5.993 | 0.025 | 2.730 | 1.224–6.261 | 0.015 |
| RCA occlusion post PCI | 2.736 | 1.104–6.715 | 0.028 | 2.647 | 1.094–6.328 | 0.029 |
| LVEF | 3.171 | 0.879–10.972 | 0.071 | 3.547 | 0.997–12.222 | 0.046 |
| NT-proBNP | 3.649 | 1.589–8.404 | 0.002 | 3.909 | 1.735–8.886 | <0.001 |
| HDL | 1.725 | 0.750–4.197 | 0.211 | – | – | – |
| MYO | 1.678 | 0.756–3.693 | 0.198 | – | – | – |

Note: CI, confidence interval. HDL, high-density lipoprotein; RCA, right coronary artery; LVEF, left ventricular ejection fraction; MYO, myoglobin; NYHA, New York Heart Association classification; *OR*, odds ratio; PCI, Percutaneous coronary intervention.

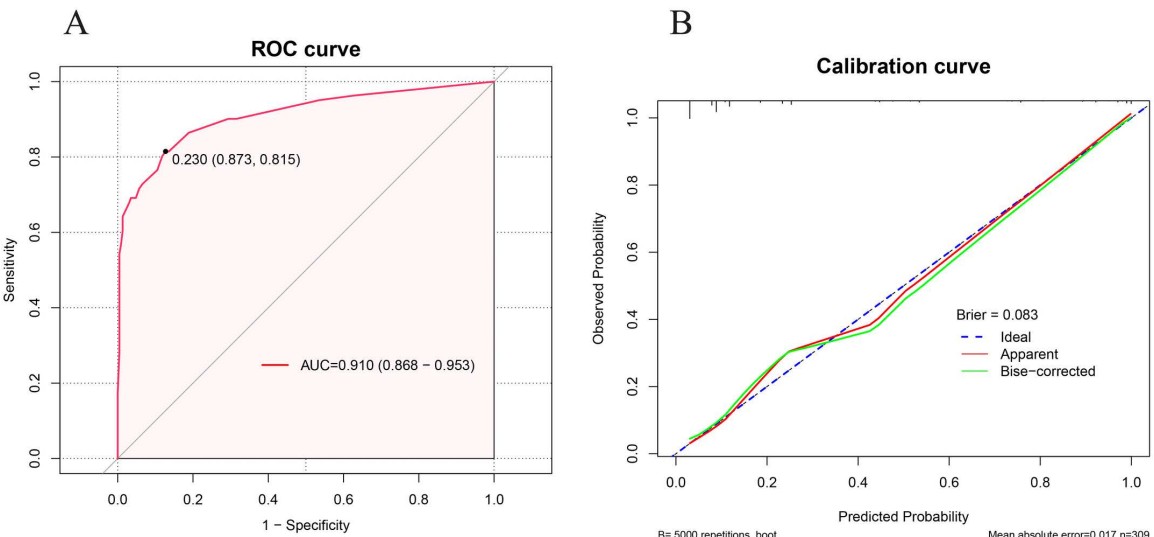

**Fig 2. Receiver operating characteristic curve (A) and calibration curve (B) of clinical prediction model.**

## 4. Discussion

This study developed a clinical prediction model to assess the risk of in-hospital HF after PCI by retrospectively collecting case data. The results showed that the model has excellent diagnostic and predictive capabilities. It includes five key predictor variables: NYHA classification, smoking status, RCA occlusion post-PCI, LVEF, and NT-proBNP levels. The nomogram derived from this model provided accurate risk predictions tailored to individual patient conditions, thereby supporting clinical decision making and improving the efficiency of treatment strategies.

Steyerberg et al. [24] proposed that when constructing clinical prediction models, it is important not only to focus on model performance, such as discrimination, but also to consider clinical applicability and convenience to promote the model's use in clinical practice. In this study, LASSO regression was used to select the most critical influencing factors. LASSO regression performs feature selection by applying L1 regularization, which compresses the regression coefficients to zero and effectively handles multicollinearity [25]. This was particularly important when considering multiple laboratory indicators. Subsequently, based on stepwise regression and model significance, high-density lipoprotein and myoglobin

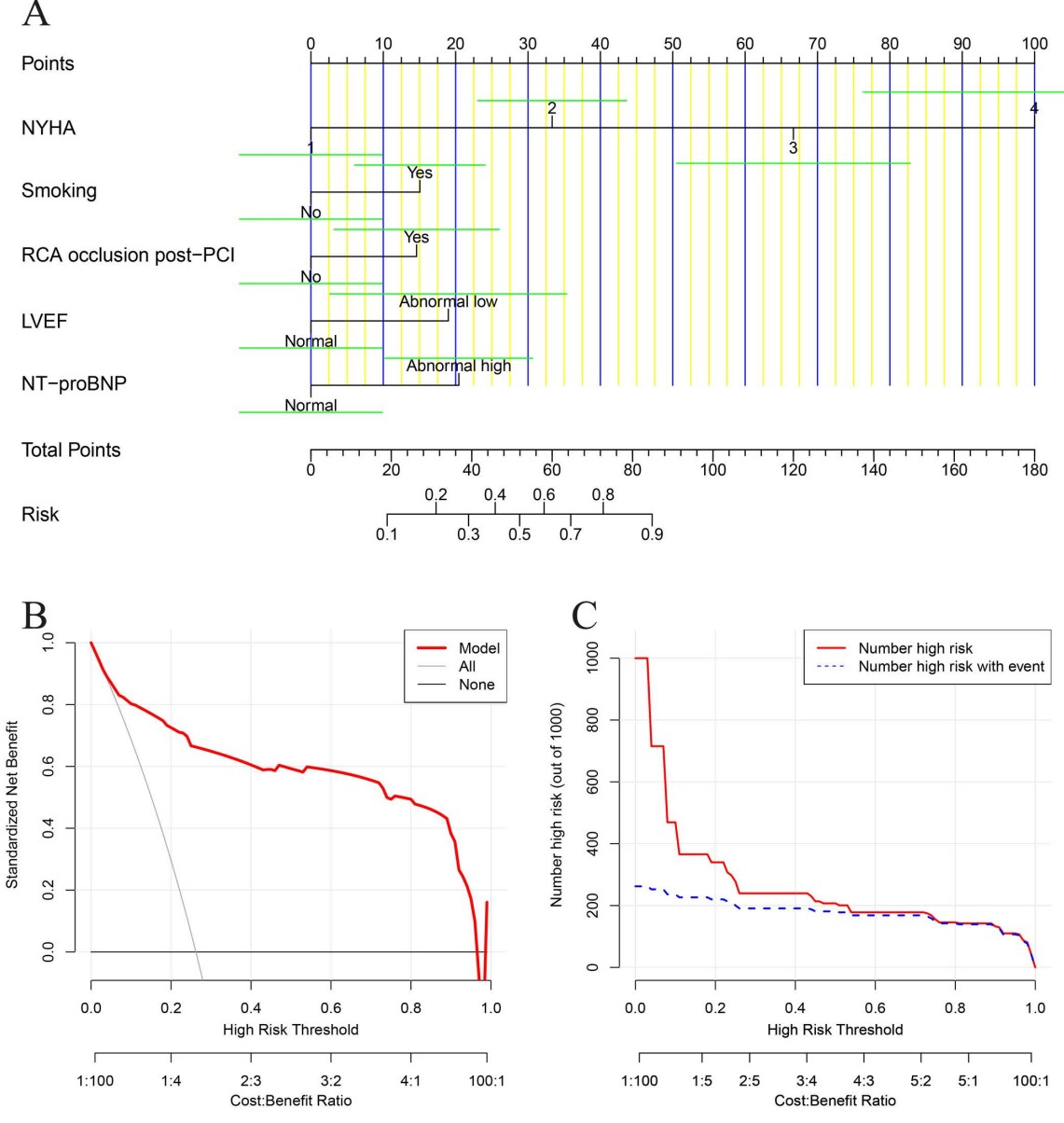

**Fig 3. Application of clinical prediction model.** (A) Nomogram, where variable levels corresponded to the "Points" axis, with the green line indicating the confidence interval; the "Total Points" reflected the predicted risk on the "Risk" axis; (B) Decision curve analysis, where "None" assumed no patients are treated (net benefit is zero) and "All" assumed all patients are treated (showing net benefit in this extreme case). "Model" represents the benefit from decisions based on the model; (C) Clinical impact curve, where "Number high risk" indicated the total high-risk patients at different thresholds, and "Number high risk with events" indicated those high-risk patients with actual events.

were excluded, further simplifying the model while ensuring the contribution of each variable and reducing its complexity, making it more applicable in clinical practice.

The NYHA classification of cardiac function, was the most important variable in our model. This classification reflects not only the cardiac function status of patients with coronary artery disease but also the severity and complexity of coronary artery lesions [26]. It serves as a foundation for the diagnosis, treatment, and prognosis of heart failure [27].

Furthermore, several studies have confirmed that smoking significantly increases the risk of angina pectoris, acute myocardial infarction, and sudden death, and it is a known risk factor for developing heart failure [28]. In a study by Ning Ding MD et al., smokers were 2.28 times more likely than non-smokers to be diagnosed with heart failure with preserved ejection fraction and 2.16 times more likely to develop heart failure with reduced ejection fraction [29]. The main components of tobacco, including carbon monoxide and nicotine, are important contributors to cardiovascular disease [30]. Carbon monoxide increases oxidative stress, leading to impaired mitochondrial function, inflammation, disrupted endothelial function, and deterioration of renal function—all closely linked to heart failure development [31]. Additionally, smoking dramatically raises systolic and diastolic blood pressure, systemic vascular resistance, pulmonary artery pressure, and pulmonary vascular resistance, all of which are recognized risk factors for heart failure [32,33].

Among the laboratory indicators, our study identified a strong association between the development of heart failure after PCI and NT-proBNP and LVEF, consistent with existing clinical guidelines. NT-proBNP is an inactive substance formed by cleaving the signal peptide from proBNP, the precursor of brain natriuretic peptide (BNP), which is transcribed into proBNP [34]. Following the onset of heart failure, hemodynamic changes activate the natriuretic peptide system, leading to increased synthesis and secretion of BNP, which subsequently elevates NT-proBNP levels. This makes NT-proBNP a widely used biomarker in diagnosing heart failure [35]. Additionally, compared to BNP, NT-proBNP has a longer and more stable half-life, enhancing its sensitivity for early heart failure diagnosis. It better reflects the new synthesis of BNP and the activation of the BNP pathway over shorter periods, minimizing the influence of external factors, making it more suitable for monitoring heart failure [36]. LVEF is an important indicator for assessing left ventricular systolic function, defined as the ratio of the volume of blood ejected with each heartbeat to the left ventricular end-diastolic volume. A decrease in ejection fraction typically indicates cardiac dysfunction and provides objective evidence for patients with heart failure [37]. As a key criterion for assessing cardiac function, ejection fraction not only aids in classifying patients with cardiac dysfunction but also holds significant importance in their prognostic evaluation [38].

Our study revealed a strong positive association between RCA occlusion after PCI and the development of heart failure. Previous studies have shown that heart failure may also be caused by dysfunction of the right ventricle, which has unique anatomical and physiological characteristics, with a thin and compliant wall that is extremely sensitive to pressure loading, and an acute elevation of afterload that can result in a substantial reduction in stroke volume. Therefore, when the right ventricle is affected by various cardiac diseases, myocardial hypertrophy, fibrosis, and metabolic abnormalities can occur, exacerbating the deterioration of right ventricular function, and these mechanical and functional changes ultimately lead to circulatory stasis and low cardiac output, which may ultimately lead to the development of right heart failure [3,39]. The severity of RCA occlusion correlates with the extent of right heart infarction [40]. Femia et al. [41] reported that RCA occlusion leads to a larger infarct size and directly impairs right ventricular systolic function, triggering an increase in right ventricular end-diastolic pressure, a significant increase in right atrial pressure, and a decrease in left ventricular preload through septal leftward motion, leading to systemic hypoperfusion. If left untreated, persistent ischemia exacerbates necrosis of cardiomyocytes, ventricular remodeling, and fibrosis, which further induce right ventricular dilatation and systolic-diastolic uncoupling. This results in dilatation and systolic-diastolic dysfunction, forming a vicious cycle of "ischemia-dysfunction-ischemia", causing chronic heart failure. Although RCA occlusion does not significantly alter cardiac output or right and left ventricular pressures, it does lead to a notable reduction in right ventricular contractility [42]. The primary function of the right ventricle is to propel blood into the pulmonary arteries; thus, decreased right heart contractility can increase pulmonary artery resistance, leading to greater pulmonary artery obstruction. This obstruction can result in decreased cardiac output and elevated right ventricular diastolic pressure, ultimately leading to right ventricular failure or systemic shock. Jiang et al. [43] conducted a retrospective study in 2024 on the prognosis of percutaneous coronary intervention for chronic total occlusion of the right coronary artery, enrolling 2,659 patients. Their findings demonstrated that the RCA has more complex lesion characteristics than the LAD and LCX, with longer lesion lengths, more tortuous vascularization, greater difficulty in opening, and higher risk of restenosis, and that prolonged ischemia can lead to impaired or

remodeled right ventricular systolic function and exacerbated right heart afterload, which may create critical conditions for the development of right heart failure.

This study further conducted subgroup analyses based on gender and age to evaluate the applicability of the model in different populations and identify potential biases. The results showed that in male and older patients, the predictive performance of the model was consistent with the main analysis, with key predictors remaining stable. However, in female and younger patients, only NYHA remained statistically significant. This gender difference may be primarily attributed to the significant imbalance in sample size between males and females, leading to male characteristics dominating the selection of predictive variables. The most direct evidence is smoking: since the proportion of smokers among female patients is inherently low, the predictive role of smoking in this group was not sufficiently demonstrated, which requires further validation in future studies. Additionally, a study by Merella et al. [15] found that over a five-year follow-up, the risk of HF was significantly higher in males than in females (*HR* = 1.22, 95% *CI*: 1.03–1.44), further supporting the necessity of developing separate predictive models for different genders. Similarly, in the age subgroup analysis, the model showed greater stability in the older age group, likely due to the higher prevalence of cardiovascular risk factors such as hypertension, diabetes, and coronary atherosclerosis, which may amplify the effects of key predictive variables. Therefore, future studies should consider developing separate predictive models for different sex and age groups to enhance model applicability and precision, ultimately optimizing individualized risk assessment and clinical decision-making.

In our study, the omission of key clinical variables and established scoring systems may have some impact on the model's performance and generalizability. First, the exclusion of lesion length, inflammatory markers, peak troponin levels, and the number of stents implanted during PCI due to excessive missing data may lead to an incomplete representation of factors influencing post-PCI heart failure. These variables are closely related to myocardial injury, procedural complexity, and systemic inflammation, and their absence may weaken the model's ability to fully capture the multifactorial nature of post-PCI heart failure risk. Second, widely used risk scores, such as the Syntax score, HEART score, TIMI risk score, Killip classification, and GRACE score, integrate multiple prognostic factors and have been externally validated in different populations. The absence of these scores may reduce the comparative utility and interpretability that these scoring systems provide. Furthermore, while we included NYHA classification as an indicator of heart failure severity, it does not account for ischemic burden, coronary complexity, or overall cardiovascular risk, which could have been better represented by the omitted scoring systems. As a result, the predictive accuracy of our model may be affected by the absence of these variables, potentially leading to an underestimation of risk in some patients.

Among related clinical model studies, the one most comparable to this study is the research by Fei Yu [17] et al. They analyzed the in-hospital mortality of patients with acute ST-segment elevation myocardial infarction who developed acute heart failure after undergoing PCI. They found that key predictive factors included LVEF and Killip class IV, which aligns with this study's identification of LVEF and NYHA classification as critical predictors. Furthermore, other clinical prediction model studies on heart failure have reported similar findings. For example, in a 10-year heart failure prediction model study conducted by Paul M Hendrik et al. [44]. involving 602 patients with moderate or complex congenital heart disease, NT-proBNP was found to significantly improve the C-statistic of the clinical prediction model. Similarly, in a clinical prediction model study by Xiyi Huang [45] on patients with coronary heart disease complicated by HF, where the primary outcome was major adverse cardiovascular events within one year, key predictive factors included NYHA classification ≥3, a history of heart failure, and NT-proBNP. Additionally, in Wenwu Tang's study [46] on predicting heart failure hospitalization and mortality in patients undergoing maintenance hemodialysis, LVEF and NT-proBNP were also identified as key predictors. These findings suggest that indicators such as LVEF, NYHA classification, and NT-proBNP hold significant predictive value across different types of heart failure patients, further confirming the validity and clinical applicability of the results of this study.

Compared with existing studies, the present model did not include certain potential predictors. This discrepancy may stem from differences in study design and population characteristics. Firstly, the D-dimer indicator included in Yu et al.'s

[17] model was excluded from the final model in this study, which may be related to differences in the time window of the study endpoints. As a marker of coagulation function, elevated D-dimer levels have been validated in multiple studies to be associated with an increased risk of acute-phase mortality [47]. However, the mechanism underlying in-hospital heart failure, the focus of this study, may be more closely related to myocardial remodeling (e.g., ventricular wall stress reflected by NT-proBNP) and hemodynamic changes (e.g., contractile function indicated by LVEF). Secondly, the prognostic model for CHD patients with acute HF developed by Huang et al. [45] incorporated historical variables such as diabetes history, whereas the present study did not retain such variables. This difference may be attributed to the strict selection criteria of the study population. Specifically, this study focused on patients with newly developed or aggravated heart failure after PCI. In this particular context, immediate postoperative indicators were found to have stronger predictive power than pre-existing medical history. This contrasts with the population in Huang's study, which included chronic heart failure patients who were more susceptible to the cumulative effects of underlying diseases due to prolonged cardiac remodeling.

However, our study has several limitations. First, the data were collected retrospectively, which may introduce recall bias. Second, both the case and control groups were drawn from the same hospital, which may introduce an admission rate bias. The characteristics of patients in this particular hospital may differ from those in other medical institutions, which may affect the generalizability of our findings. Future studies should aim to select healthy controls from a wider community to improve the representativeness of the results and ensure greater external validity. Third, the relatively small sample size of our study may limit the statistical validity and credibility of the results. Small sample sizes can lead to increased variability, reduced ability to detect true effects, and misleading conclusions. In addition, the gender distribution in our sample was uneven, with a higher proportion of male participants than female participants. This gender difference may further limit the generalizability of the findings, especially in understanding gender differences in the outcomes of interest. Therefore, future studies should increase the sample size and ensure a more balanced gender distribution to improve the reliability, statistical validity, and applicability of the findings. Fourth, it is important to include data from different regions and hospitals to effectively address issues of population heterogeneity. By increasing the diversity of the sample, we can improve the applicability of the clinical prediction model. Fifth, although many potential predictors were considered in this study, some important variables were not included, such as lesion length, inflammatory markers, peak troponin levels, and number of stents implanted during PCI. These factors are strongly associated with the development of HF and may have an important impact on the accuracy and completeness of prediction models. Future studies should build on this study by further incorporating these key variables and considering other possible potential predictors, such as genetic factors, lifestyle and comorbidities, to more comprehensively assess the risk of heart failure. Sixth, previous studies have indicated that the characteristics of patients with HFpEF, HFrEF, and HFmrEF differ significantly. For example, in comparison with HFpEF subjects, patients with HFrEF were younger, more commonly male, and more likely to have an ischaemic etiology and left bundle branch block. The HFmrEF group resembled the HFrEF group in some features but had less left ventricular and atrial dilation [48]. This highlighted the necessity of refining clinical prediction models for different subgroups. However, due to the limited sample size of this study, in particular the small number of patients with HFrEF and HFmrEF, we were unable to investigate in depth the potential differences between these subgroups. This limitation may affect our understanding of the pathophysiological mechanisms and clinical characteristics between different heart failure phenotypes. Future studies should expand the sample size, especially in patients with HFrEF and HFmrEF, to more fully analyze the unique characteristics and prognostic differences between these subgroups. Seventh, the occurrence of heart failure is not limited to the inpatient setting; it can also manifest after discharge, such as within 30 days or even longer post-discharge. The cross-sectional nature of this study restricted our ability to assess long-term outcomes. Future studies should adopt a longitudinal design to better evaluate the long-term outcomes and influencing factors related to HF. Eighth, the model developed in this study has only been validated internally. While internal validation provides initial evidence of the model's performance, it may not fully capture its generalizability to different populations or settings. External validation using datasets from other hospitals or regions is crucial to assess the model's robustness, reproducibility, and applicability

in diverse clinical environments. Ninth, some widely used scoring systems such as Syntax score, HEART score, TIMI risk score, Killip classification and GRACE score were not included in this study. These scoring systems are valuable in predicting the risk of cardiovascular events and heart failure and may provide additional information on the predictive power of the model. Failure to include these scoring systems may have limited the comprehensiveness of the model and comparability with other studies. Future studies should consider incorporating these scoring systems to further optimize the predictive ability and clinical applicability of the model.

## 5. Conclusion

This study developed a clinical prediction model for assessing in-hospital heart failure risk after PCI in acute coronary artery disease patients. Key variables in the model included NYHA classification, smoking, RCA occlusion post-PCI, NT-proBNP, and LVEF. The model demonstrated excellent diagnostic efficacy, showing strong consistency and discrimination. However, while these findings suggested potential clinical utility, further studies are needed to confirm its real-world applicability. Moreover, external validation is essential to assess the model's generalizability, and further refinement is required to optimize its performance across different genders and age subgroups. It can help identify high-risk individuals likely to develop heart failure during hospital admission after PCI and may serve as a basis for guiding personalized prevention and treatment.

## Supporting information

**S1 Table. Normal value range of laboratory indicators.**
(DOCX)

**S2 Table. Regression coefficients from multivariable logistic analysis of HF risk after PCI in ACS patients.**
(DOCX)

**S3 Table. Multivariable logistic regression analysis stratified by gender and age.**
(DOCX)

**S1 Data. Data file of study participant information.**
(CSV)

## Acknowledgments

The authors would like to give their special thanks to the school and the tutor for their support and assistance.

## Author contributions

**Conceptualization:** Zhenlian Ning, Bin Huang.

**Data curation:** Zhenlian Ning, Bing Li, Ziming Ning, Beili Zhu, Mengfan Zhao.

**Formal analysis:** Zhenlian Ning, Bing Li, Ziming Ning.

**Investigation:** Ziming Ning, Beili Zhu.

**Methodology:** Zhenlian Ning, Bing Li, Ziming Ning.

**Project administration:** Bing Li, Ziming Ning, Mengfan Zhao.

**Resources:** Zhenlian Ning, Bing Li, Mengfan Zhao.

**Software:** Ziming Ning, Beili Zhu, Mengfan Zhao.

**Supervision:** Zhenlian Ning, Bin Huang.

**Validation:** Zhenlian Ning, Bing Li, Ziming Ning, Bin Huang.

**Visualization:** Zhenlian Ning, Bin Huang.

**Writing – original draft:** Zhenlian Ning, Bing Li, Ziming Ning.

**Writing – review & editing:** Zhenlian Ning, Bing Li, Bin Huang.

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
