## [Decision Letter · Decision Letter 0]

28 Jan 2025

PONE-D-24-52362Development and Validation of a Clinical Prediction Model for In-Hospital Heart Failure Risk Following PCI in Patients with Coronary Artery DiseasePLOS ONE

Dear Dr. Huang,

Thank you for submitting your manuscript to PLOS ONE. After careful consideration, we feel that it has merit but does not fully meet PLOS ONE’s publication criteria as it currently stands. Therefore, we invite you to submit a revised version of the manuscript that addresses the points raised during the review process.

We look forward to receiving your revised manuscript.

Kind regards,

Arturo Cesaro, MD

Academic Editor

PLOS ONE

**Journal Requirements:**

3. In the online submission form, you indicated that your data is available only on request from a third party. Please note that your Data Availability Statement is currently missing the contact details for the third party, such as an email address or a link to where data requests can be made. Please update your statement with the missing information. 

Reviewers' comments:

Reviewer's Responses to Questions

**Comments to the Author**

1. Is the manuscript technically sound, and do the data support the conclusions?

Reviewer #1: Yes

Reviewer #2: Yes

2. Has the statistical analysis been performed appropriately and rigorously? 

Reviewer #1: Yes

Reviewer #2: Yes

3. Have the authors made all data underlying the findings in their manuscript fully available?

Reviewer #1: No

Reviewer #2: Yes

4. Is the manuscript presented in an intelligible fashion and written in standard English?

Reviewer #1: Yes

Reviewer #2: Yes

5. Review Comments to the Author

**Reviewer #1: ** Feedback for author(s):

The authors presented a well-written manuscript titled "Development and Validation of a Clinical Prediction Model for In-Hospital Heart Failure Risk Following PCI in Patients with Coronary Artery Disease." The manuscript focuses on developing and validating a prediction model to identify high-risk patients who may develop in-hospital heart failure following percutaneous coronary intervention (PCI) in acute coronary syndrome (ACS) patients.

General comments

Strengths:

Clinical Relevance: The topic is highly relevant, given the significant risks of heart failure following PCI in ACS patients. The model could have a strong impact on clinical decision-making.

Robust Methodology: The manuscript employs well-established statistical techniques such as LASSO regression and logistic regression, and it includes model validation through ROC curves and calibration analysis.

Nomogram Utility: The final clinical prediction model in the form of a nomogram provides a practical tool for clinicians.

Weaknesses:

Sample Size: The study includes a relatively small sample (309 patients), which may limit the generalizability of the findings.

Lack of External Validation: The model has only been validated internally. External validation using datasets from other hospitals would increase the model’s robustness.

Exclusion of Key Variables: Important factors like lesion length, inflammatory markers, and peak troponin levels were not considered in the analysis, which may affect the comprehensiveness of the prediction model.

Heart Failure Subgroups: While the study acknowledges different types of heart failure (HFpEF, HFrEF, and HFmrEF), the small sample size prevents meaningful exploration of these subgroups, which could be a key limitation.

Specific Comments

Introduction:

The introduction provides a solid background on the importance of predicting heart failure risk after PCI. However, it could be enhanced by referencing recent studies that discuss the relationship between coronary artery revascularization and left ventricular dysfunction. Specifically, I suggest incorporating "Understanding the role of coronary artery revascularization in patients with left ventricular dysfunction and multivessel disease" ([https://doi.org/10.1007/s10741-023-10335-0]) to further elaborate on the complex interplay between coronary revascularization and heart failure outcomes, especially in patients with multivessel disease and left ventricular dysfunction.

Materials and Methods:

The inclusion and exclusion criteria are well defined, but the absence of certain variables such as lesion length and inflammatory markers could have been explained further. Including a rationale for not including these factors would improve the transparency of the methodology. Additionally, more details about the bootstrap validation process used for model evaluation would strengthen the description of the internal validation process.

Results:

The results are clearly presented with appropriate statistical analysis. However, the manuscript could benefit from a deeper exploration of the predictors in the context of specific heart failure subtypes (e.g., HFrEF, HFpEF). While the paper mentions these subgroups, the small sample sizes in each category prevent further analysis, which could be addressed in future research.

Discussion:

The discussion is well-rounded and provides a clear interpretation of the findings. However, it could be enhanced by comparing the study’s results to other existing predictive models. A discussion on how this model compares to other models used to predict heart failure risk in PCI patients could provide a clearer perspective on its strengths and limitations. The paper should also acknowledge the lack of external validation and the limitations associated with a small sample size.

Conclusion:

The conclusion appropriately highlights the practical implications of the clinical prediction model. A suggestion for future work, particularly the inclusion of more external data and further validation of the model in different patient populations, would be beneficial to enhance the model’s generalizability

Final comments:

**Reviewer #2: ** Dear Editor, this study; The abstruct seems sufficient in terms of introduction, results and discussion areas. It was a fluent and interesting work.

The issues that I see as lacking and that he wants to criticize are as follows.

1- A slightly larger number of patients would have increased the generalizability and quality of the study.

2- It would be better if the number of men and women were close to each other.

3- I see it as a major deficiency that several scoring systems were not included in the study. For example; Such as Syntax score, Heart score, TIMI risk score, Killip classification, Grace score.

Finally, I liked the work, but I think it needs revision, keeping my concerns above.

Kind regards

6. PLOS authors have the option to publish the peer review history of their article (what does this mean? ). If published, this will include your full peer review and any attached files.

**Do you want your identity to be public for this peer review?** For information about this choice, including consent withdrawal, please see our Privacy Policy .

Reviewer #1: No

Reviewer #2: **Yes: ** Azmi Eyiol

---

## [Author Response · Author response to Decision Letter 1]

13 Feb 2025

Dear Editor and Reviewers,

Thank you very much for your valuable feedback on our manuscript. After carefully reviewing the comments, we recognize that the previous version of the manuscript had several shortcomings that may have affected its overall quality. We assure you that this research has been conducted with the utmost academic rigor and in strict compliance with scholarly standards. At the same time, we deeply appreciate the patience and professionalism you have shown throughout the review process.

There is no doubt that your comments are of great value in revising and improving our manuscript. We have thoroughly revised the manuscript based on your suggestions, including providing additional details on the experimental methods, reanalyzing the data, and further refining the discussion section. Please find our point-by-point responses and the revised manuscript in the uploaded files. All line numbers referenced below correspond to the revised version.

Once again, we sincerely thank you and all the reviewers for your valuable advice and support.

Yours sincerely,

Bin Huang

Respond to Reviewer #2:

Reviewer #2: Dear Editor, this study; The abstruct seems sufficient in terms of introduction, results and discussion areas. It was a fluent and interesting work.

1. To improve the generalizability and quality of the study, it would be beneficial to increase the sample size and achieve a more balanced gender distribution (with a closer number of men and women).

Response: Thank you for your careful review of our paper. We appreciate your pointing out the possible unequal gender distribution and the small sample size of the population covered in the study. We have addressed this point in the limitations section of the manuscript (see lines 310-318). In our future research, we will pay more attention to the gender balance of the sample, try to increase the sample size, optimize the sample structure, establish cooperation with more medical institutions, expand the sample collection channels, and strive to collect more samples in a wider area to further improve our study. Thank you again for your comments, which will help us to continuously improve and enhance the quality of our study.

2. I see it as a major deficiency that several scoring systems were not included in the study. For example; Such as Syntax score, Heart score, TIMI risk score, Killip classification, Grace score.

Response: Thank you for pointing out the critical issue that multiple scoring systems were not included in this study. The study was designed to investigate the key predictive factors influencing the development of heart failure after PCI and their clinical utility, and thus focused on the included factors or indicators such as patient baseline characteristics, procedure-related markers, and laboratory findings, without comprehensive consideration of systems such as the Syntax score, HEART score, TIMI risk score, Killip classification, and GRACE score. . However, we are aware of these circumstances, as highlighted in the limitations section of this study (see lines 352-360). Nevertheless, we fully recognize the important value of these scoring systems and their non-inclusion is indeed a major drawback. For follow-up studies, we plan to work with more clinical organizations to fully incorporate these scoring systems for reassessment. Thank you again for your professional review, it means a lot to improve the quality of the study.

Respond to Reviewer #1:

Reviewer #1: The authors presented a well-written manuscript titled "Development and Validation of a Clinical Prediction Model for In-Hospital Heart Failure Risk Following PCI in Patients with Coronary Artery Disease." The manuscript focuses on developing and validating a prediction model to identify high-risk patients who may develop in-hospital heart failure following percutaneous coronary intervention (PCI) in acute coronary syndrome (ACS) patients.

General comments

Strengths:

Clinical Relevance: The topic is highly relevant, given the significant risks of heart failure following PCI in ACS patients. The model could have a strong impact on clinical decision-making.

Robust Methodology: The manuscript employs well-established statistical techniques such as LASSO regression and logistic regression, and it includes model validation through ROC curves and calibration analysis.

Nomogram Utility: The final clinical prediction model in the form of a nomogram provides a practical tool for clinicians.

Weaknesses:

Sample Size: The study includes a relatively small sample (309 patients), which may limit the generalizability of the findings.

Lack of External Validation: The model has only been validated internally. External validation using datasets from other hospitals would increase the model’s robustness.

Exclusion of Key Variables: Important factors like lesion length, inflammatory markers, and peak troponin levels were not considered in the analysis, which may affect the comprehensiveness of the prediction model.

1. The study includes a relatively small sample (309 patients), which may limit the generalizability of the findings.

Response: Thank you very much for your careful review of our paper. We strongly agree with the points you raise about the small sample size of the study population. We have addressed this point in the limitations section of the paper (see lines 310-318). In our future research, we will try to increase the sample size and broaden the channels of sample collection to further improve our study.

2. The model has only been validated internally. External validation using datasets from other hospitals would increase the model’s robustness.

Response: Thank you very much for pointing out the critical issue that the model is only internally validated, your feedback means a lot to our research. We have addressed this in the limitations section (see lines 347-352) of the paper manuscript.

3. Important factors like lesion length, inflammatory markers, and peak troponin levels were not considered in the analysis, which may affect the comprehensiveness of the prediction model.

Response: Thank you for pointing out that important factors such as lesion length, inflammatory markers, and peak troponin levels were not included in the analysis, and we fully agree that this may have affected the comprehensiveness and accuracy of the predictive model. However, we apologize for the fact that we were unable to find the other information you mentioned related to lesion severity, such as lesion length, in the hospital information system, which limited our use of this information. We have therefore highlighted this point in the limitations section of the text to give the reader a fuller understanding of the potential shortcomings of this study, as detailed in lines 321 to 329. We are aware of the importance of these factors to the model and will take active steps to improve them. In future studies, we will communicate with multiple hospitals to expand the scope of the data collection, and once the data collection is complete, we will evaluate the association between the new factors and the model, screen out the significant factors to be included in the model, re-train and optimize the model, and use a variety of metrics to evaluate and analyze them in comparison to the original model. We will certainly do our best to improve the study and enhance the quality of the model. Thanks again for your valuable comments.

4. While the study acknowledges different types of heart failure (HFpEF, HFrEF, and HFmrEF), the small sample size prevents meaningful exploration of these subgroups, which could be a key limitation.

Response: Thank you for pointing out the main limitation that the small sample size affected the subgroup analysis. In the current study, the sample size was not as large as desired due to the limited number of patients who met the inclusion criteria in a given time period and the difficulty in accessing healthcare resources in some areas. We found that the number of patients with heart failure with moderately reduced ejection fraction (HFmrEF) and heart failure with reduced ejection fraction (HFrEF) was too small (n = 22) to adequately fit the model, and we therefore abandoned this approach; however, we greatly appreciate your suggestions for optimizing and improving the predictive model to expand its applicability in different patient subgroups. Indeed, this is one of the priorities for clinical predictive modeling research. We plan to implement your ideas by increasing the sample size in future studies. We have emphasized this in the limitations section of this study (see lines 336 - 342). Thank you again for your constructive comments and patience in reviewing our manuscript.

5. The introduction provides a solid background on the importance of predicting heart failure risk after PCI. However, it could be enhanced by referencing recent studies that discuss the relationship between coronary artery revascularization and left ventricular dysfunction. Specifically, I suggest incorporating "Understanding the role of coronary artery revascularization in patients with left ventricular dysfunction and multivessel disease" ([https://doi.org/10.1007/s10741-023-10335-0]) to further elaborate on the complex interplay between coronary revascularization and heart failure outcomes, especially in patients with multivessel disease and left ventricular dysfunction.

Response: Thank you for your careful review of our study and your valuable suggestions. Your suggestion to cite recent studies that discuss the relationship between coronary revascularization and left ventricular dysfunction was very valuable. We fully agree that citing "Understanding the role of coronary artery revascularization in patients with left ventricular dysfunction and multivessel disease" will help to further clarify the relationship between coronary revascularization and heart failure. revascularization and the complex interactions between heart failure outcomes, especially in patients with multivessel disease and left ventricular dysfunction. We will include a discussion of this study in the Introduction section to enhance the depth and breadth of the Background section, as described in lines 46 to 51.

6. More details about the bootstrap validation process used for model evaluation would strengthen the description of the internal validation process.

Response: Thank you for your careful review and valuable comments on our paper. For more details on the bootstrap validation process you mentioned, we have described it in lines 151 to 156.

7. The results are clearly presented with appropriate statistical analysis. However, the manuscript could benefit from a deeper exploration of the predictors in the context of specific heart failure subtypes (e.g., HFrEF, HFpEF). While the paper mentions these subgroups, the small sample sizes in each category prevent further analysis, which could be addressed in future research.

Response: Thank you for your comment. As addressed in our response to comment #4, the small sample sizes limited subgroup analysis of HFrEF and HFpEF. This is noted in the limitations section (lines 336-342). Future studies will expand the sample size to enable deeper exploration of heart failure subtypes. We appreciate your valuable suggestion.

8. The discussion is well-rounded and provides a clear interpretation of the findings. However, it could be enhanced by comparing the study’s results to other existing predictive models. A discussion on how this model compares to other models used to predict heart failure risk in PCI patients could provide a clearer perspective on its strengths and limitations. 

Response: We sincerely thank the reviewers for their valuable feedback and constructive suggestions, which helped strengthen the discussion section. We agree that a comparison of our model with other existing prediction models can more clearly demonstrate its strengths and limitations. In response to this comment, we have expanded the discussion section, see lines 283-302. We believe that these additions further enhance the quality of the paper by providing a broader context for the results and demonstrating the clinical relevance of our model.

---

## [Decision Letter · Decision Letter 1]

25 Mar 2025

PONE-D-24-52362R1Development and Validation of a Clinical Prediction Model for In-Hospital Heart Failure Risk Following PCI in Patients with Coronary Artery DiseasePLOS ONE

Dear Dr. Huang,

Thank you for submitting your manuscript to PLOS ONE. After careful consideration, we feel that it has merit but does not fully meet PLOS ONE’s publication criteria as it currently stands. Therefore, we invite you to submit a revised version of the manuscript that addresses the points raised during the review process.

We look forward to receiving your revised manuscript.

Kind regards,

Arturo Cesaro, MD

Academic Editor

PLOS ONE

Journal Requirements:

Reviewers' comments:

Reviewer's Responses to Questions

**Comments to the Author**

1. If the authors have adequately addressed your comments raised in a previous round of review and you feel that this manuscript is now acceptable for publication, you may indicate that here to bypass the “Comments to the Author” section, enter your conflict of interest statement in the “Confidential to Editor” section, and submit your "Accept" recommendation.

Reviewer #1: All comments have been addressed

Reviewer #2: All comments have been addressed

2. Is the manuscript technically sound, and do the data support the conclusions?

Reviewer #1: Partly

Reviewer #2: Yes

3. Has the statistical analysis been performed appropriately and rigorously? 

Reviewer #1: No

Reviewer #2: Yes

4. Have the authors made all data underlying the findings in their manuscript fully available?

Reviewer #1: No

Reviewer #2: Yes

5. Is the manuscript presented in an intelligible fashion and written in standard English?

Reviewer #1: Yes

Reviewer #2: Yes

6. Review Comments to the Author

Reviewer #1: Major Weaknesses (Unresolved Concerns):

Sample Size and Population Bias: The small sample (n=309), with nearly 80% male participants, limits statistical power and generalizability. While the limitation is acknowledged, the authors should provide further analysis stratified by gender and discuss implications more explicitly.

Exclusion of Important Predictors: Key clinical predictors such as lesion length, peak troponin, inflammatory markers, and number of stents were omitted. The reasons provided are understandable, but their clinical impact on prediction performance must be critically discussed, particularly for lesion severity and troponin as established HF predictors.

Absence of Comparator Risk Scores: No comparison was made with widely used scoring systems such as GRACE, TIMI, Killip, or SYNTAX scores. Reviewer #1 clearly requested this. Without benchmarking against these models, it is difficult to judge the added value of the proposed tool.

Incomplete Outcome Definition: The criteria used to diagnose in-hospital HF post-PCI, especially whether standardized guidelines or clinical judgment were used, remain insufficiently detailed.

Detailed Comments:

Introduction:

The cited literature on coronary revascularization and LV dysfunction is relevant. However, the rationale for developing a new model—rather than modifying existing scores—needs better justification.

Methods:

Provide a clearer operational definition of heart failure diagnosis.

Specify how missing data were managed and whether any variable interactions were explored.

Include a rationale for excluding key variables (e.g., lesion length, troponin) in the Methods section.

Results:

Provide stratified model performance by gender or age to assess potential bias.

Include the full regression coefficients in supplementary materials.

Discussion:

Expand on how the model compares (or could compare) with existing models.

Critically evaluate the omission of key clinical variables and scoring systems.

Strengthen the rationale for including each of the five final predictors.

Clarify how RCA occlusion impacts right heart failure risk based on literature.

Conclusion:

Temper statements about clinical application.

Emphasize the need for external validation and model refinement.

Reviewer #2: Although this article is interesting, beautifully and fluently written, it focuses on a subject that needs to be developed with future studies. In general, it is a study that I liked.

7. PLOS authors have the option to publish the peer review history of their article (what does this mean? ). If published, this will include your full peer review and any attached files.

**Do you want your identity to be public for this peer review?** For information about this choice, including consent withdrawal, please see our Privacy Policy .

Reviewer #1: No

Reviewer #2: **Yes: ** Azmi Eyiol

---

## [Author Response · Author response to Decision Letter 2]

13 Apr 2025

Dear Editor and Reviewers,

Thank you for your constructive feedback and the opportunity to revise our manuscript. We sincerely appreciate the time and expertise invested by the reviewers, particularly Reviewer #1’s detailed comments, which have significantly strengthened the methodological rigor and clarity of our work.

Key Improvements in the Revised Manuscript:

All concerns raised by Reviewer #1 have been thoroughly addressed, with methodological rigor enhanced and discussions expanded.

Reviewer #2 had no further concerns, confirming the validity of our findings.

Detailed point-by-point responses to each comment are provided in the attached file. The revised manuscript and supplementary materials are also uploaded for your convenience.

We hope these revisions meet the journal’s standards and welcome any further suggestions to improve the manuscript.

Yours sincerely,

Bin Huang

Respond to Reviewer #1:

Major Weaknesses:

Sample Size and Population Bias: The small sample (n=309), with nearly 80% male participants, limits statistical power and generalizability. While the limitation is acknowledged, the authors should provide further analysis stratified by gender and discuss implications more explicitly.

Exclusion of Important Predictors: Key clinical predictors such as lesion length, peak troponin, inflammatory markers, and number of stents were omitted. The reasons provided are understandable, but their clinical impact on prediction performance must be critically discussed, particularly for lesion severity and troponin as established HF predictors.

Absence of Comparator Risk Scores: No comparison was made with widely used scoring systems such as GRACE, TIMI, Killip, or SYNTAX scores. Reviewer #1 clearly requested this. Without benchmarking against these models, it is difficult to judge the added value of the proposed tool.

Incomplete Outcome Definition: The criteria used to diagnose in-hospital HF post-PCI, especially whether standardized guidelines or clinical judgment were used, remain insufficiently detailed.

Detailed Comments:

1. The cited literature on coronary revascularization and LV dysfunction is relevant. However, the rationale for developing a new model—rather than modifying existing scores—needs better justification.

Response: Thank you for the comment. We have revised the introduction to better clarify the aim of our study. The model developed by Yu et al. focused on predicting in-hospital mortality in patients who had already developed heart failure after PCI. In contrast, our model is designed to predict the occurrence of new-onset heart failure during hospitalization following PCI. Our model aims to help identify high-risk patients early and support timely clinical intervention (Lines 56–61).

2. Provide a clearer operational definition of heart failure diagnosis.

Response: In response to your valuable feedback regarding potential gaps in diagnostic criteria, we have updated and refined the heart failure diagnostic framework by synthesizing evidence from the 2021 ESC and 2022 ACC/AHA consensus guidelines. The revised criteria systematically incorporate clinical manifestations (symptoms and signs), objective biomarkers, imaging parameters, and exclusionary diagnoses, while establishing a dynamic assessment process that integrates symptom evaluation, physical examination findings, biomarker analysis, and imaging studies (Lines 100–114).

3. Specify how missing data were managed and whether any variable interactions were explored.

Response: We are very grateful to the reviewers for their valuable comments. We generated five imputed datasets through 100 iterations of the multivariate imputation by chained equations (MICE) method (Lines 183–185).

4 Include a rationale for excluding key variables (e.g., lesion length, troponin) in the Methods section.

Response: Thank you very much for your valuable comments. A large amount of missing data could likely introduce biases in the analysis. Thus, to ensure the reliability of the results, we decided to exclude them (Lines 149–152).

5.Provide stratified model performance by gender or age to assess potential bias.

Response: Thank you for your instruction. We've addressed the need to provide stratified model performance by gender and age. Table S3 in the manuscript shows the model results across these subgroups (Lines 1623–165, Lines 211–215).

6.Include the full regression coefficients in supplementary materials.

Response: Thanks for the instruction. We've added the full regression coefficients to the supplementary materials as required. You can find them in Table S3.

7.Expand on how the model compares (or could compare) with existing models.

Response: We sincerely accept your suggestion of the need for model comparison. In the original manuscript, we mainly described the similarities between our model and existing ones. Following your advice, we have now added a more detailed comparison of the differences in predictor variables, which may provide readers with a more comprehensive understanding of our model’s unique contributions (Lines 392–410).

8. Critically evaluate the omission of key clinical variables and scoring systems.

Response: Thank you for your insightful comment on the omission of key variables and scoring systems. In our initial study, due to extensive missing data, we overlooked crucial variables like lesion length. The absence of widely-used scores, such as the Syntax score, also limited our study. In the revised manuscript, we have objectively evaluated the implications of these omissions and highlighted our study's limitations (Lines 355–371).

9. Strengthen the rationale for including each of the five final predictors.

Response: Thank you for your careful review and the valuable comment. We selected the five predictors by considering both statistical significance and clinical applicability. These variables are easy to obtain in practice and may enhance the model’s usefulness in real-world clinical application. The relevant reasons have been elaborated in the article (Lines 254–264).

10.Clarify how RCA occlusion impacts right heart failure risk based on literature.

Response: I truly appreciate your astute question. When the RCA is occluded, it directly disrupts the blood supply to the right ventricle. This disruption can lead to reduced oxygen and nutrient delivery, subsequently weakening the right ventricular function and significantly increasing the susceptibility to right heart failure. In the revised manuscript, we have elaborated it (Lines 301–334).

11.The authors should provide further analysis stratified by gender and discuss implications more explicitly.

Response: Thank you for the suggestion. We have added subgroup analyses by gender and age in the revised manuscript. The model performed well in males and older patients, while performance was limited in females and younger patients, likely due to sample size imbalance. We also discussed the implications and supported the need for future models tailored to different subgroups (Lines 335–354).

12.Temper statements about clinical application and emphasize the need for external validation and model refinement.

Response: We sincerely appreciate your suggestions. We have emphasized the importance of external validation and the need for further model refinement in the revised manuscript. In addition, we have revised the overall conclusion of the study to be more cautious, reflecting the preliminary nature of the findings. (Lines 471–475).

---

## [Decision Letter · Decision Letter 2]

7 May 2025

Development and Validation of a Clinical Prediction Model for In-Hospital Heart Failure Risk Following PCI in Patients with Coronary Artery Disease

PONE-D-24-52362R2

Dear Dr. Huang,

We’re pleased to inform you that your manuscript has been judged scientifically suitable for publication and will be formally accepted for publication once it meets all outstanding technical requirements.

Kind regards,

Arturo Cesaro, MD

Academic Editor

PLOS ONE

Additional Editor Comments (optional):

Please observe the word count and citation style. For further details, please consult the Instructions for Authors on the website.

Reviewers' comments:

Reviewer's Responses to Questions

**Comments to the Author**

1. If the authors have adequately addressed your comments raised in a previous round of review and you feel that this manuscript is now acceptable for publication, you may indicate that here to bypass the “Comments to the Author” section, enter your conflict of interest statement in the “Confidential to Editor” section, and submit your "Accept" recommendation.

Reviewer #1: All comments have been addressed

Reviewer #2: All comments have been addressed

2. Is the manuscript technically sound, and do the data support the conclusions?

Reviewer #1: Yes

Reviewer #2: Yes

3. Has the statistical analysis been performed appropriately and rigorously? 

Reviewer #1: Yes

Reviewer #2: Yes

4. Have the authors made all data underlying the findings in their manuscript fully available?

Reviewer #1: Yes

Reviewer #2: Yes

5. Is the manuscript presented in an intelligible fashion and written in standard English?

Reviewer #1: Yes

Reviewer #2: Yes

6. Review Comments to the Author

Reviewer #1: No further comments.

All the comments have been addressed. I recommend the acceptance of the manuscript.

Reviewer #2: Dear author,

Thank you for considering and responding to the corrections and suggestions requested by all reviewers. Your work will shed light on this field. You have produced a beautiful study and publication.

7. PLOS authors have the option to publish the peer review history of their article (what does this mean? ). If published, this will include your full peer review and any attached files.

**Do you want your identity to be public for this peer review?** For information about this choice, including consent withdrawal, please see our Privacy Policy .

Reviewer #1: No

Reviewer #2: **Yes: ** Azmi Eyiol

---

## [Editor Report · Acceptance letter]

PONE-D-24-52362R2

PLOS ONE

Dear Dr. Huang,

I'm pleased to inform you that your manuscript has been deemed suitable for publication in PLOS ONE. Congratulations! Your manuscript is now being handed over to our production team.

Kind regards,

on behalf of

Dr. Arturo Cesaro

Academic Editor

PLOS ONE